# Advertising ultra-processed foods around urban and rural schools in Kenya

Caroline H. Karugu[1,2]*, Gershim Asiki[1,3], Milkah N. Wanjohi[1], Veronica Ojiambo[1,4], Sharon Mugo[1], Peter M. Kaberia[5], Richard E. Sanya[1], Amos Laar[6], Michelle Holdsworth[7], Stefanie Vandevijvere[8], Charles Agyemang[2,9]

**1** Chronic Diseases Management Unit, African Population Health Research Center, Nairobi, Kenya, **2** Department of Public and Occupational Health, Amsterdam Public Health, University of Amsterdam Medical Centers, Amsterdam, The Netherlands, **3** Department of Women's and Children's Health, Karolinska Institute, Stockholm, Sweden, **4** Department of Public & Global Health, University of Nairobi, Kenya, **5** West Africa Regional Office, African Population and Health Research Center, Dakar, Senegal, **6** School of Public Health, University of Ghana, Accra, Ghana, **7** UMR MoISA (Montpellier Interdisciplinary Centre On Sustainable Agri-Food Systems), (Univ Montpellier, CIRAD, CIHEAM-IAMM, INRAE, Institut Agro, IRD), Montpellier, France, **8** Sciensano, Service of Lifestyle and Chronic Diseases, Brussels, Belgium, **9** Division of Endocrinology, Diabetes, and Metabolism, Department of Medicine, Johns Hopkins University School of Medicine, Baltimore, Maryland, United States of America

\* ckarugu@aphrc.org

## Abstract

Marketing of ultra-processed foods (UPFs) can influence children's food preferences and consumption patterns. However, limited data exist on the extent and nature of UPF marketing around schools in low- and middle-income countries, including Kenya. This study assessed the extent, type, and content of food and beverage advertising near schools in urban and rural settings in Kenya. We conducted a cross-sectional study in June–July 2021 across three Kenyan counties—Nairobi (urban), Mombasa (coastal urban), and Baringo (rural). Each county was stratified by socioeconomic status (SES), and schools were randomly selected. Food and beverage advertisements within a 250-meter radius of schools were documented. Data collected included the type of product, location, and promotional techniques used. Advertised products were categorized using the NOVA classification and the INFORMAS framework. Descriptive statistics were used to summarize advertisement patterns, and Poisson regression was applied to identify factors associated with UPF advertising. A total of 2,300 food and beverage advertisements were documented around 500 schools. Urban areas had a higher median number of advertisements (median = 25, IQR: 25–160) compared to rural areas (median = 10, IQR: 4–13). Nearly 48% of all advertisements featured UPFs. The most frequent promotional strategy involved cartoon and company-owned characters, while price discounts were the most common premium offers. In multivariate analysis, Baringo County showed a higher rate of UPF advertisements compared to Nairobi (PRR: 1.17, 95% CI: 1.01–1.36), as did lower versus higher SES areas (PRR: 1.10, 95% CI: 1.01–1.20). UPFs

**Data availability statement:** The dataset of this study is available at the African Population and Health Research Center's Microdata portal (https://microdataportal.aphrc.org/index.php/catalog/174). The data request will undergo data sharing agreement policy review and guidelines at the African Population and Health Research Centre.

**Funding:** This research was funded by The International Development Research Centre (IDRC), Canada, to the African Population and Health Research Centre (APHRC) and The SA MRC Centre for Health Economics and Decision Science /PRICELESS, (Grant number: 1019132-001 to GA). The funders had no role in study design, data collection and analysis, decision to publish, or preparation of the manuscript.

**Competing interests:** The authors have declared that no competing interests exist.

are commonly advertised around schools in Kenya, often using strategies that appeal to children. Regulatory efforts are needed to limit the marketing of unhealthy foods in school environments.

## Introduction

Overweight and obesity are a growing global challenge largely driven by food environments and the consumption of unhealthy foods, as well as insufficient physical activity [1]. Obesity prevalence is challenge projected to increase over time, particularly in low and middle-income countries (LMICs) [2]. A major challenge in LMICs is the double burden of malnutrition in children. Overweight and obesity and undernutrition in childhood are all associated with an increased future risk of non-communicable diseases in adulthood [3].

The economic and industrial transition in various settings has substantially impacted the food environments that children and adults are exposed to. There is an increasing trend in the marketing of unhealthy foods such as ready-to-eat snacks and sweetened drinks, and beverages targeting children and adolescents [4–6]. Children are progressively exposed to unhealthy foods and beverages by industries targeting settings and platforms such as schools, the traditional media, including TV and radio, and social media [3,7,8]. Several persuasive techniques, such as cartoons, company-owned characters, celebrity brand endorsements, gifts and collectibles, price discounts, and game application downloads, are used to entice children toward the purchase and consumption of unhealthy foods [9,10]. Marketing of unhealthy foods influences dietary habits, attitudes, and preferences toward unhealthy foods, and increased consumption of the marketed food and beverages [6]. In both high-income countries (HICs) and LMICs, there is an increasing trend in advertising unhealthy foods and beverages targeting children [11].

A range of studies from both low- and middle-income countries (LMICs) and high-income countries (HICs) have documented the pervasive promotion of unhealthy food products directed at children and adolescents [12–18]. Studies conducted in New Zealand and Australia, representing different socioeconomic deprivation areas, showed that the majority of the advertised food and beverage products were prohibited by the World Health Organization (WHO) nutrition profile model and were mostly soft drinks and alcoholic drinks [12,13]. In the Philippines and Guatemala, studies assessing outdoor advertisements around school neighborhoods showed a high density of advertisements around schools and a high level of advertisements of unhealthy foods that target children [19,20]. A study conducted in Uganda's capital, Kampala, around school neighborhoods showed that more than 80% of advertisements were for sugar-sweetened beverages, especially in urban settings [21]. In Ghana, approximately 69% of the food and beverage products advertised were in the ultra-processed foods (UPFs) category, with cartoon and company-owned characters as the main promotional characters used [18]. Most of the studies on marketing around school environments have focused on urban neighborhoods [18,21]. This study addresses a

critical gap by identifying variations in the marketing of unhealthy foods in child-centered environments across both urban and rural settings, thereby informing more context-specific policy recommendations. Furthermore, such findings will inform policy that focuses on LMIC urban areas and in rural settings where industrialization is progressively taking root.

In this study, we assessed the extent, content, type, and power of advertising of ultra-processed/unhealthy foods around schools in both urban and rural settings in Kenya. We further explored other factors associated with the marketing of UPFs.

## Materials and methods

### Ethics statement

The study protocol was approved by the AMREF-Health Ethics and Scientific Review Committee in Kenya (ESRC/P901/2020) and the National Commission for Science, Technology (NACOSTI- P/22/19104). Permission and community mobilization were carried out through engagement with school heads and sub-county level administrators.

### Study design

This was a cross-sectional study conducted in Kenya from June 2021 to July 2021, which assessed the food and beverage marketing around primary and secondary schools. The study adopted the International Network for Food and Obesity/NCDs Research Monitoring and Action Support (INFORMAS) methodology, the outdoor food advertisement protocol in particular [22].

### Study setting

Three counties (Nairobi, Mombasa, and Baringo) were selected conveniently to represent different contexts including urban and rural, low and high socio-economic status (SES) within urban settings, and a tourist destination in Kenya. Each county was stratified into low and high SES sub-counties using poverty-level data from the 2019 Kenya National Bureau of Statistics (KNBS) estimates [23]. In Nairobi County Westlands (higher SES), Langata (higher SES), Embakasi (lower SES), Kibra (lower SES), and Mathare (lower SES) were selected. In Mombasa county, Mvita (Higher SES), and Kisauni (lower SES) sub-counties were selected. Lastly, in Baringo county, we included Mogotio (higher SES), and Baringo North (lower SES) sub-counties.

### Sample size and sampling

The list of schools in Nairobi, Mombasa, and Baringo counties by location (county, subcounty), level, and type was provided by the Ministry of Education (MOE), Kenya. First, we estimated a sample size of 500 schools, considering a prevalence of 23% of unhealthy food advertisements in low socioeconomic areas [24], a power of 80%, and an Alpha error set to 5%. We further adjusted this sample size considering a 10% non-response rate in all the settings. We employed a multistage stratified sampling technique in this study. The 500 schools were further randomly allocated proportionately to the number of schools by level (primary or secondary and type (private or public) in each county and sub-county using the stratified sampling method.

### Data collection procedures

After obtaining ethical approvals from the AMREF-Health Ethics and Scientific Review Committee in Kenya (approval number: ESRC/P901/2020) and permission from the school heads, the research team, comprising two data collectors per school at a time, was deployed to the selected schools. They were trained to identify the required distance/radius from the main school gates and entrances and collect data on the advertisements within the defined radius [24], as per the INFORMAS outdoor advertisements protocol used in previous studies in LMIC settings (18, 22).

## Food marketing assessments/ study variables

The details of the schools, including the school name, school type (private, public, primary school, secondary school), school setting (urban, rural, peri-urban), school population, and number of school entrances, were recorded. All the advertisements within the pre-determined Euclidean radius around the schools were recorded. This involved first identifying the location of the food and beverage advertisement; whether the product was placed in food shops, buildings, on the roadside, bus shelter/bus stage, or mobile carts. The format of advertisements (billboards, posters/banners, standing signs, painted walls and buildings, digital signs, and store merchandising) was also recorded. The size of the advertisements was categorized as small (> A4 but < 1.3 m × 1.9 m), medium (≥ 1.3 m × 1.9 m but < 2.0 m × 2.5 m), or large (≥ 2.0 m × 2.5 m). Further, we assessed the content of the advertisement (logos/company pictorials, text, combined logos, and text) and the number of food products in the individual advertisement (single-food product type, two-food product type, three-food product type, and >3 food product type). Each food item advertisement was recorded, along with any associated promotional characters and premium offers, such as giveaways, discounts, or collectable items. The promotional characters include cartoon or company-owned characters, famous sports persons, movie titles, or images with kids. The premium offers included price discounts, limited stock editions, and buy one, get one free.

## Food classification

The food and beverage products advertised around school neighborhoods were categorized based on the INFORMAS food categories [25], and the NOVA classification systems [18]. Alcoholic beverages were not included in the food categorizations and were reported as a separate group. The INFORMAS food categorization classifies foods into core (healthy) and non-core (unhealthy) food categories [18,25–26]. Examples of core foods are fruits and vegetables, while non-core foods are unhealthier food options such as sugar-sweetened beverages. The NOVA food classification system classifies foods into i) unprocessed/minimally processed which are foods in their raw state that have undergone no industrialization processes and have no added sugars and salts ii) processed culinary ingredients which are mainly additives to foods products such as cooking oils, iii) processed foods which contains added nutrients such as salts and sugars, and iv) Ultra-processed foods(UPFs) which are highly processed foods that have undergone multiple industrialization processes and contains high levels of sugar, salts and fats [27]. Additionally, the NOVA food categories were reclassified into a binary variable to facilitate analysis: ultra-processed foods (UPFs) versus non-UPFs. This binary distinction was based on the NOVA classification, where group IV (ultra-processed foods) was categorized as UPFs, and groups I to III comprising unprocessed/minimally processed foods, processed culinary ingredients, and processed foods were grouped as non-UPFs. This reclassification allowed for clearer comparisons of the extent of UPF marketing relative to all other food types.

## Statistical analysis

The analysis was guided by the INFORMAS protocol [27]. Descriptive statistics were used to show the distribution of foods advertised by counties, SES areas, and other characteristics such as school types and level, with the degree or extent of urbanisation. We used medians and interquartile ranges to describe the extent of advertisements around schools. The outcome of the study was the extent of advertisements of core (healthy) and non-core (unhealthy) foods, and UPFs' advertisements around schools. The predictor variables included counties, SES levels, rural/urban settings, school type, school population, and the gender of the student population within the schools. The multivariable Poisson regression models were used to determine the association between the count of food and beverages advertised around the schools and other characteristics, such as SES status and school type. We adjusted for clustering within the individual sub-counties in the Poisson regression models to account for the differences in the settings of these regions. Deviance and Pearson chi-square post-estimation tests were employed to assess the goodness-of-fit of the models. Stata Version 18 was used for Statistical analysis, and the significance level was set at $\alpha = 0.05$.

## Results

### School characteristics description

Advertising was assessed around all 500 schools: Nairobi county schools (n = 215, 43.0%), Baringo county schools (n = 173, 34.6%), and Mombasa county schools (n = 112, 22.4%). More than 50% of the schools were public schools, while 78.4% were primary schools (Table 1). Nearly two-thirds of the schools were from urban settings, while most schools (>90%) were mixed gender (both male and female pupils). According to the SES assessment of the schools, 60.0% of them were in the lower SES (Table 1). The Median (Interquartile Range (IQR)) school population in all the counties was 335(176, 500) pupils. The schools had entrances ranging from 1 to 8.

### Extent of advertisements around schools

We observed a total of 2300 food and beverage advertisements, most of them from Nairobi county (n = 1256; 54.8%), followed by Mombasa county (n = 878; 38.2%), and the fewest from Baringo county (n = 166; 7.2%). According to the density of advertisements around schools, the food and beverage products were more advertised in Mombasa (Median = 25, IQR = 12, 78) and Nairobi counties (Median = 25, IQR = 6, 69), and lower in Baringo county (Median = 8, IQR = 5, 14) (Table 2). We observed a higher number of food and beverage advertisements in urban areas (Median = 30, range = 25, 160), in

**Table 1. School characteristics in Nairobi, Mombasa, and Baringo Counties, Kenya in 2021.**

| Variable | Description | Frequency (Percentage) (N = 500) |
|---|---|---|
|  |  | n (%) |
| **County** |  |  |
|  | Nairobi | 215 (43.0) |
|  | Mombasa | 112 (22.4) |
|  | Baringo | 173 (34.6) |
| **School ownership** |  |  |
|  | Private school | 238 (47.6) |
|  | Public school | 262 (52.4) |
| **School type** |  |  |
|  | Primary school | 392 (78.4) |
|  | Secondary school | 108 (21.6) |
| **School setting** |  |  |
|  | Urban | 323 (64.6) |
|  | Rural | 177 (35.4) |
| **Gender of pupils in the schools** |  |  |
|  | Boys school | 20 (4.0) |
|  | Girls school | 15 (3.0) |
|  | Mixed school | 465 (93.0) |
| **School population** |  |  |
|  | Median (IQR) | 335(176, 500) |
| **School SES area** |  |  |
|  | Lower SES | 300 (60.0) |
|  | Higher SES | 200 (40.0) |
| **Number of school entrances** |  |  |
|  | Mean (SD), range | 1.3 (0.8), (1, 8) |

**Global Public Health**

PLOS

**Table 2. Median number of food and beverage advertisements around a 250 m radius around schools in Kenya by county, SES levels, and school type in Nairobi, Mombasa, and Baringo counties in 2021.**

| Variable | Characteristics | Number of schools (N = 500) | Median advertisements around schools | (25th, 75-percentiles) |
|---|---|---|---|---|
| **County** | | | | |
| | Nairobi county | 215 | 25 | (12,78) |
| | Mombasa county | 112 | 25 | (7, 69) |
| | Baringo county | 173 | 8 | (5, 15) |
| **School setting** | | | | |
| | Urban | 323 | 30 | (25, 160) |
| | Rural | 177 | 10 | (4, 13) |
| **SES level** | | | | |
| | Lower SES | 300 | 22 | (11, 83) |
| | Higher SES | 200 | 25 | (12, 69) |
| **School type** | | | | |
| | Private primary | 190 | 19 | (15, 92) |
| | Public primary | 202 | 16 | (8, 35) |
| | Private Secondary | 48 | 10 | (4, 14) |
| | Public secondary | 60 | 6 | (3, 15) |

higher SES areas (Median = 25, IQR = 12, 69), and around private primary schools (Median = 19, IQR = 15, 92) (Table 2**).** We found a lower median number of advertisements around private secondary schools and public secondary schools.

## Features of food and beverage advertisements

Table 3 presents the distribution characteristics of food and beverage advertisements across the surveyed counties. A significant majority (85.5%) of the advertisements were located within food retail stores, while 5.8% were displayed on buildings and 5.4% along roadways. Posters and banners constituted the predominant format of display (60.6%), followed by in-store merchandising (25.4%) and painted buildings or walls (8.4%).

Single-product promotions accounted for 61.8% of all advertisements, whereas 16.0% featured a combination of more than three food product types. In terms of content, 86.7% of the advertisements included a combination of logos, pictorial elements, and text. Regarding advertisement size, 46.5% were categorized as small, while 38.2% and 15.2% were medium- and large-sized, respectively.

## Types of foods and beverages advertised using INFORMAS and NOVA food categories

Table 4 presents the INFORMAS classification of food and beverage advertisements. Overall, 46.0% of the advertisements featured unhealthy food options, 46.4% promoted healthy foods, and approximately 5% fell into a miscellaneous category. Advertisements for alcoholic beverages accounted for 3% of the total. Across all counties, the most frequently advertised food category was sugar-sweetened beverages, classified as unhealthy, which represented over 33.1% of all food and beverage advertisements. Among healthy food categories, meat and meat alternatives were the most commonly advertised, comprising 12% of all advertisements.

In Nairobi County, 44.4% of the advertisements promoted unhealthy foods (median = 37.5, IQR = 16.5–47.5), while 46.7% featured healthy foods (median = 79, IQR = 56–114) (Tables 4 and 5). In Mombasa County, healthy food advertisements accounted for 49.8% of the total, while unhealthy foods made up 44.8%. In Baringo County, unhealthy foods comprised approximately 63.9% of food and beverage advertisements (median = 15, IQR = 53–72) (Tables 4 and 5).

Sugar-sweetened beverages remained the most advertised category across all three counties, contributing to approximately 34.2%, 27.5%, and 54.8% of food and beverage advertisements in Nairobi, Mombasa, and Baringo counties,

**Table 3. Techniques for food advertising in/around schools in Nairobi, Mombasa, and Baringo Counties in 2021.**

| Description | Frequency (%) | |
|---|---|---|
| **Setting of advertisement** | | |
| Food shop | 1,966 | (85.5%) |
| Road | 125 | (5.4%) |
| Building | 133 | (5.8%) |
| Bus shelter/ bus stage/bus stop/train | 2 | (0.1%) |
| Mobile cart/stall or vending machine | 17 | (0.7%) |
| Other specify [a] | 57 | (2.5%) |
| **Format of advertisement** | | |
| Billboard | 86 | (3.7%) |
| Poster or banner | 1,393 | (60.6%) |
| Standing sign | 30 | (1.3%) |
| Painted building/wall | 192 | (8.4%) |
| Digital signs/ LED | 10 | (0.4%) |
| Store merchandising | 583 | (25.4%) |
| Others specify [b] | 6 | (0.3%) |
| **Number of product types in advertisement** | | |
| None (only company/brand mentioned) | 82 | (3.6%) |
| 1 food product type | 1,422 | (61.8%) |
| 2 food product types | 210 | (9.1%) |
| 3 food product types | 219 | (9.5%) |
| > 3 food product types | 367 | (16.0%) |
| **Content of the advertisement** | | |
| Logos/Pictorial | 111 | (4.8%) |
| Text | 195 | (8.5%) |
| Combined logos, pictorials, and text | 1994 | (86.7%) |
| **Size of advertisement** | | |
| Small (> A4 but < 1.3 m × 1.9 m), | 1070 | (46.5%) |
| Medium (≥ 1.3 m × 1.9 m but < 2.0 m × 2.5 m) | 878 | (38.2%) |
| Large (≥ 2.0 m × 2.5 m) | 352 | (15.2%) |

Notes: [a] Other settings include; advertisements on gates, bars, restaurants, and liquor shops.

[b] Other formats include: Branded moving cars and calendar adverts.

respectively (Table 4). Bivariate analyses revealed significant differences in the promotion of core and non-core foods by county, socioeconomic setting, school location (urban vs. rural), and school type.

According to the NOVA food categorization, nearly equal proportions of the advertisements were for UPFs versus unprocessed/minimally processed (49% versus 48%) in all the counties (Fig 1). In Mombasa, unprocessed/minimally processed foods were advertised more (52%) than UPF, while in Baringo, 64% of the advertisements were UPF.

### Description of promotional/marketing strategies in the advertisements

62.3% of the advertisements had no promotional characters present. Among those with promotional characters, cartoon or company-owned characters were the most frequently observed promotional strategy (37.2%). Other observed promotional methods accounting for 2.0% of the advertisements included famous sports persons (n = 2), sports events (n = 4), movie titles (n = 2), and advertisements for children, e.g., images of a child (n = 14). The most common premium offers in the

**Table 4. Food categories and distribution by the proportion of food advertisements in/ around schools in Nairobi, Mombasa, and Baringo Counties in 2021.**

| INFORMAS Food Categories | All (N=2300) # | % | Nairobi (N=1256) # | % | Mombasa (N=878) # | % | Baringo (N=166) # | % |
|---|---|---|---|---|---|---|---|---|
| **Core (Healthy) Foods** | **1067** | **(46.4%)** | **586** | **(46.7%)** | **437** | **(49.8%)** | **44** | **(26.5%)** |
| Breads, cereals, rice and rice products without added fat, sugar or salt, noodles | 228 | (9.9%) | 112 | (8.9%) | 99 | (11.3%) | 17 | (10.2%) |
| Fruits and fruit products without added fats, sugars, or salt | 92 | (4.0%) | 41 | (3.3%) | 45 | (5.1%) | 6 | (3.6%) |
| Vegetables and vegetable products without added fats, sugars, or salt | 141 | (6.1%) | 71 | (5.7%) | 68 | (7.7%) | 2 | (1.2%) |
| Milks and yoghurts (≤3g fat/100g), cheese (≤15g fat/100g), and their alternatives | 179 | (7.8%) | 149 | (11.9%) | 25 | (2.9%) | 5 | (3.0%) |
| Meat and meat alternatives - include meat, poultry, fish, legumes, eggs and raw unsalted nuts | 281 | (12.2%) | 116 | (9.2%) | 165 | (18.8%) | – | – |
| Oils high in mono- or polyunsaturated fats, and low-fat savoury sauces (<10g fat/100g) | 30 | (1.3%) | 18 | (1.4%) | 5 | (0.6%) | 7 | (4.2%) |
| Bottled water (including unflavoured mineral and soda waters) | 116 | (5.0%) | 79 | (6.3%) | 30 | (3.4%) | 7 | (4.2%) |
| **Non-Core (unhealthy) Foods** | **1057** | **(46.0%)** | **558** | **(44.4%)** | **393** | **(44.8%)** | **106** | **(63.9%)** |
| Sweetbreads, cakes, muffins, sweet buns, sweet biscuits, pies and pastries | 148 | (6.4%) | 54 | (4.3%) | 79 | (9.0% | 15 | (9.0%) |
| Meat and meat alternatives processed or preserved in salt | 25 | (1.1%) | 9 | (0.7%) | 16 | (1.8%) | – | – |
| Sweet snack foods | 18 | (0.8%) | 14 | (1.1%) | 4 | (0.5%) | – | – |
| Savory snack foods (added salt or fat) | 38 | (1.7%) | 24 | (1.9%) | 14 | (1.6%) | – | – |
| High fat/salt meals - frozen or packaged meals (>6g saturated fat/serve, >900mg sodium/serve) | 67 | (2.9%) | 28 | (2.2%) | 39 | (4.4%) | – | – |
| Sugar-sweetened drinks - include soft drinks, sweetened tea drinks | 761 | (33.1%) | 429 | (34.2%) | 241 | (27.5%) | 91 | (54.8%) |
| **Miscellaneous foods** | **109** | **(4.7%)** | **65** | **(5.2%)** | **28** | **(3.2%)** | **16** | **(9.6%)** |
| Recipe additions (including soup cubes, oils, dried herbs, and seasonings) | 19 | (0.8%) | 14 | (1.1%) | 4 | (0.5%) | 1 | (0.6%) |
| Tea and coffee (excluding sweetened powder-based teas or coffees) | 90 | (3.9%) | 51 | (4.1%) | 24 | (2.7%) | 15 | (9.0%) |
| **Alcohol** | **67** | **(2.9%)** | **47** | **(3.7%)** | **20** | **(2.3%)** | **–** | **–** |

advertisements were price discounts (n=49) and limited editions, e.g., last stock (n=33). Other premium offers observed include: contests (n=6), buy one get one free (n=5), gift or collectibles (n=5), and 20% extra (n=4).

### Factors associated with UPF advertisements around schools

There was a significantly higher level of advertisements of UPFs in Baringo County (prevalence rate ratios (PRR): 1.28, 95% CI: 1.01-1.20) compared to urban Nairobi County (Table 6). We observed a significantly higher level of UPFS advertisements in Lower SES settings (PRR: 1.10, 95% CI: 1.01-1.20).

## Discussion

We observed a high level of marketing of UPFs and other unhealthy food and beverages around schools in Kenya, with nearly half of all the food advertisements rated as unhealthy. There was a substantial strategic placement of advertisements within a 250m radius around schools, with the highest numbers being observed in the urban counties (Nairobi

**Table 5. Healthiness of food and beverage categories advertised by school characteristics in Kenya in 2021.**

| Variable | Characteristics | Core (healthy) foods<br>Median (25th, 75 percentile) | Non-core (unhealthy) foods<br>Median (25th, 75 percentile) | Miscellaneous<br>Median (25th, 75 percentile) | P-value |
|---|---|---|---|---|---|
| **County** | | | | | **0.000** |
| | Nairobi county | 79(56, 114) | 37.5(16.5, 47.5) | 32.5(23.25, 41.75) | |
| | Mombasa county | 45(26.25, 68.0) | 20(14.5, 39.0) | 14(9.0, 19.0) | |
| | Baringo county | 7(5.25, 7.0) | 15(53, 72) | 8(4.5, 11.5) | |
| **School setting** | | | | | **0.000** |
| | Urban | 136(99, 191) | 67(31, 101.5) | 46(31.5, 60.5) | |
| | Rural | 5(4, 5) | 12(6.5, 49.5) | 8.5(5.25, 11.75) | |
| **SES level** | | | | | **0.000** |
| | Lower SES | 73(37.5, 113.5) | 14(13.5, 52.5) | 29(20, 38) | |
| | Higher SES | 62(45, 103.5) | 26(18, 68.5) | 25.5(16.75, 34.25) | |
| **School type** | | | | | **0.000** |
| | Private primary | 95(63, 127.5) | 35(22.5, 56.0) | 25.5(17.75, 33.25) | |
| | Public primary | 25(17, 30.5) | 10(5, 35) | 17.5(11.25, 23.75) | |
| | Private secondary | 12(8.5, 15.5) | 11(4.75, 14.25) | 5.5(3.25, 7.75) | |
| | Public secondary | 11(8, 25.5) | 5(3, 7) | 6(4.5, 7.5) | |

Notes: P-Value from chi-square tests for checking differences in proportions of food categories by different variables such as SES levels, Bold: Significant associations.

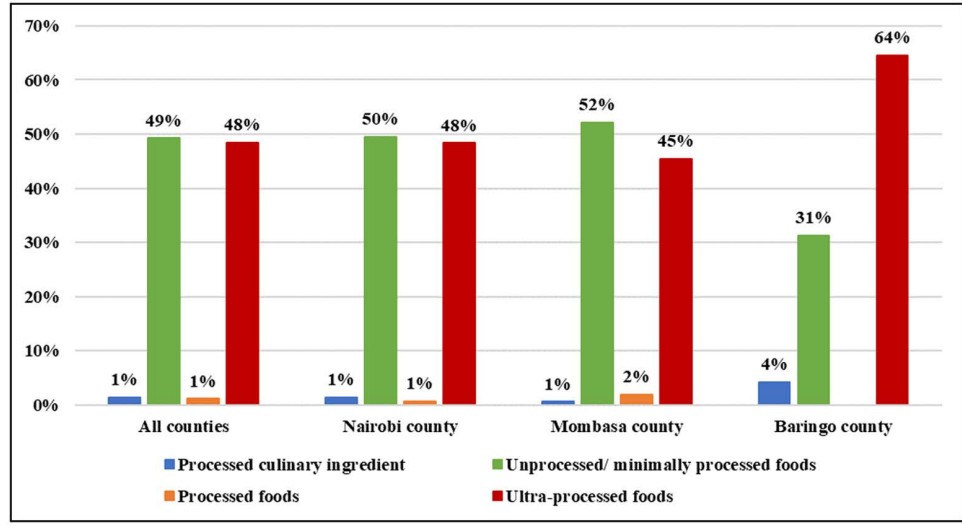

**Fig 1. County-Level Comparison of Food Advertisements by NOVA Processing Category Around Schools in Kenya.**

and Mombasa) and around primary schools. This is consistent with findings from studies in other countries where there were varying densities of advertisements by SES locations and in urban areas compared to peri-urban neighborhoods [14,18,21]. Our findings identified a significantly high level of advertisements in lower-SES areas in the various counties, which is similar to a study conducted in Ghana using the same methodology [18]. These findings show that urban settings and both lower and higher SES locations are targeted by food and beverage industries.

**Table 6. Multivariate Poisson regression output for factors associated with levels of advertisements of ultra-processed foods and beverages in Nairobi, Mombasa, and Baringo counties.**

| | PRR | 95% CI | P-value |
|---|---|---|---|
| **County** | | | |
| Ref: Nairobi county | 1 | 1 | 1 |
| Baringo county | 1.28 | [1.00-1.63] | **0.048** |
| Mombasa county | 0.99 | [0.90-1.09] | 0.872 |
| **SES level** | | | |
| Ref: High SES | 1 | 1 | 1 |
| Lower SES | 1.10 | [1.01-1.20] | **0.035** |
| **School Setting** | | | |
| Ref: Urban | 1 | 1 | 1 |
| Rural | 1.05 | [0.83-1.34] | 0.676 |
| **School type** | | | |
| Ref: Primary school | 1 | 1 | 1 |
| Secondary schools | 1.04 | [0.94-1.15] | 0.451 |

Notes: *PRR- Prevalence rate ratios, CI- Confidence Interval; SES- Socioeconomic status, Bold: Significant associations.*

Nearly half of the advertisements around schools were in the UPF/unhealthy food category, showing a substantial exposure of children to unhealthy foods, a similar trend reported elsewhere [12,18,21,28]. Different studies from LMICs have consistently shown that most advertisements for unhealthy foods strategically target children [12,13,15,29]. Additionally, we observed a significantly higher proportion of UPFs around schools in rural Baringo County compared to the other counties. Despite the lower overall number of advertisements around schools in this region, the majority of these advertisements were for UPFs. This suggests that the marketing and availability of UPFs have penetrated rural areas, where the majority of advertised products are unhealthy foods. These child-directed advertisements of unhealthy foods are a great public health concern as they increase the consumption of unhealthy foods, which will consequently increase the risk of nutrition-related NCDs [4,30,31].

The most commonly advertised unhealthy food in this study was sugar-sweetened drinks, followed by sweetened bread, cakes, and savory snacks. Most studies conducted around schools have shown a high level of promotion of sugar-sweetened beverages to children [18,21]. Studies conducted in urban Uganda and Ghana using the INFORMAS methods showed a high level of promotion of unhealthy foods, with sugar-sweetened beverages being the most advertised product [18,21]. The high level of exposure to unhealthy food advertisements creates significant risks of poor dietary choices and consequential health-harming implications for children [4,32,33].

Surprisingly, 3% of the advertisements were for alcohol, which is restricted from advertising to children in Kenya [34]. Alcohol advertisements around schools have also been observed in Uganda and Ghana, and these advertisements are associated with an increased likelihood of consumption of alcohol by children [18,21,35].

The majority of the advertisements in this study were strategically placed in the food shops/kiosks around the schools and were mostly in the form of posters and banners. Most children purchase both food and non-food products from shops around school, implying that they are exposed to advertisements for unhealthy foods. These findings are consistent with the study conducted in Ghana, where the majority of advertisements around schools were mainly placed in food shops [18]. Modern retail outlets and shops are associated with increased exposure and prominence of unhealthy food categories, and make different products and brands easily accessible for purchase by children [18,36,37].

Posters and banners are a marketing strategy used to make clear messaging about products, which can be viewed by the target market. As observed in previous studies, physical advertisements using posters and banners improve the

proximity and awareness of the food items being advertised and influence the purchase and consumption of these products [18,38,39]. This study shows the diversified marketing landscape and its strategic target to vulnerable populations such as children, stressing the need for regulations encompassing the marketing strategies, placement, content, and type of advertisements exposed to different populations. The most common promotional strategies used in this study were the cartoon or company-owned characters, hence improving the appeal of the advertisements to children. Some studies have shown that the use of cartoon and company-owned characters and mascots enhanced the purchase and consumption of unhealthy foods, and consequential poor health outcomes [9,10].

## Policy implications

This study examines the marketing of ultra-processed foods (UPFs) in the vicinity of schools across diverse socioeconomic settings in Kenya. The promotion and marketing of unhealthy foods can influence the perception of the product and thus increase the purchase intention of children or better pester power to caregivers. Consequently, this leads to the consumption of unhealthy foods associated with poor health outcomes such as overweight/obesity, and nutrition-related non-communicable disease among children and adults. The marketing of unhealthy foods, specifically targeting children, has been widespread around school neighborhoods. The marketing strategies have been improved and mirror the methods used in HICs, as the industrialization and modernization of food environments take their roots in LMICs.

This study contributes to the knowledge of the landscape assessment of marketing around school neighborhoods, with a comparison between rural and urban areas of Kenya. The most worrying observation is the targeting of younger children in primary schools, rural schools, and schools in low SES urban areas. This study calls for a multisectoral approach toward the regulation of unhealthy food advertising in school settings, regardless of their location and socio-economic status. There is a need to sensitize school authorities about the dangers of advertising unhealthy foods to children to stimulate local restrictions on such advertisements by the schools. Furthermore, it is important to note that there were substantial levels of advertisements of healthier food options in these settings. This highlights the need for regulations that strike a balance between the costs of healthier food options and the incentives for their promotion around schools. At the school level, continuous advocacy, sensitization, and education on predatory marketing of unhealthy foods and the health-harming implications should be done to counter the marketing campaigns in school neighborhoods. There needs to be a great focus by policymakers on the high level of advertisements of sugar-sweetened beverages and alcoholic beverages around schools, as elucidated in this study. The rights-based approach can be implemented as recommended by WHO, which advises that environments where children gather should be free from the marketing of unhealthy foods [40].

## Strengths and limitations

This is the first study in Kenya assessing marketing in neighborhoods where children gather, hence making it novel. This study is unique in that it entails findings of marketing around school neighborhoods from both rural and urban regions in an LMIC setting. Most marketing assessment studies have been conducted in HICs and urban areas of a few countries in Africa. Hence, the study shows a comparative assessment of the Kenyan urban and rural contexts, showing a high penetration of UPFs advertisements targeting children. The main limitation of the study was that it was conducted during the COVID-19 period, with numerous restrictions/lockdowns, which might have impacted the number of advertisements during the study period. However, most of the advertisements displayed before COVID-19 were available during the survey. So, it is possible that we did not miss a substantial number of advertisements during the study period.

## Conclusion

The findings from this study illustrate the high level of exposure to unhealthy foods and beverages targeting children in both urban and rural settings in Kenya. There is a concerning observation on the use of innovative techniques of marketing around schools, and the widespread advertisement of sugar-sweetened beverages, especially in rural settings. There

is an urgent need for the formulation of national policies on the regulation of the marketing of unhealthy foods to children in school environments. Targeted awareness and sensitization are needed at the school level to inform the children and teachers about the existing marketing of unhealthy foods targeting children.

## Supporting information

**S1 Table. NOVA food categories by different factors.**
(DOCX)

## Acknowledgments

The authors acknowledge the Ministries of Health and Education in Kenya (MOH and MOE), and the county governments of Nairobi, Mombasa, and Baringo for supporting the research in the country and in specific counties. We also acknowledge the research assistants who supported data collection.

## Author contributions

**Conceptualization:** Gershim Asiki, Milkah N. Wanjohi, Richard E. Sanya, Amos Laar, Michelle Holdsworth, Stefanie Vandevijvere, Charles Agyemang.

**Data curation:** Caroline H. Karugu, Milkah N. Wanjohi, Veronica Ojiambo, Sharon Mugo.

**Formal analysis:** Caroline H. Karugu, Peter M. Kaberia, Stefanie Vandevijvere.

**Funding acquisition:** Gershim Asiki, Amos Laar, Michelle Holdsworth, Stefanie Vandevijvere.

**Investigation:** Gershim Asiki, Milkah N. Wanjohi, Veronica Ojiambo, Sharon Mugo, Richard E. Sanya, Stefanie Vandevijvere, Charles Agyemang.

**Methodology:** Caroline H. Karugu, Gershim Asiki, Veronica Ojiambo, Peter M. Kaberia, Amos Laar, Michelle Holdsworth, Charles Agyemang.

**Project administration:** Caroline H. Karugu, Gershim Asiki, Milkah N. Wanjohi, Veronica Ojiambo.

**Resources:** Gershim Asiki, Richard E. Sanya, Amos Laar, Michelle Holdsworth, Stefanie Vandevijvere, Charles Agyemang.

**Software:** Caroline H. Karugu.

**Supervision:** Gershim Asiki, Richard E. Sanya, Amos Laar, Michelle Holdsworth, Stefanie Vandevijvere, Charles Agyemang.

**Validation:** Caroline H. Karugu, Gershim Asiki, Veronica Ojiambo, Richard E. Sanya, Amos Laar, Michelle Holdsworth, Stefanie Vandevijvere, Charles Agyemang.

**Visualization:** Caroline H. Karugu, Peter M. Kaberia.

**Writing – original draft:** Caroline H. Karugu.

**Writing – review & editing:** Gershim Asiki, Milkah N. Wanjohi, Veronica Ojiambo, Peter M. Kaberia, Richard E. Sanya, Amos Laar, Michelle Holdsworth, Stefanie Vandevijvere, Charles Agyemang.

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
