## [Decision Letter · Decision Letter 0]

PGPH-D-24-02106

Advertising ultra-processed foods around urban and rural schools in Kenya

Dear Dr. Karugu,

Thank you for submitting your manuscript to PLOS Global Public Health. After careful consideration, we feel that it has merit but does not fully meet PLOS Global Public Health’s publication criteria as it currently stands. Therefore, we invite you to submit a revised version of the manuscript that addresses the points raised during the review process.

There are few comments from the reviewers that need to be address before proceeding.

We look forward to receiving your revised manuscript.

Kind regards,

Leonor Guariguata, MPH, PhD

Academic Editor

Journal Requirements:

Additional Editor Comments (if provided):

Reviewers' comments:

Reviewer's Responses to Questions

**Comments to the Author**

1. Does this manuscript meet PLOS Global Public Health’s publication criteria ? Is the manuscript technically sound, and do the data support the conclusions? The manuscript must describe methodologically and ethically rigorous research with conclusions that are appropriately drawn based on the data presented.

Reviewer #1: Yes

Reviewer #2: Partly

2. Has the statistical analysis been performed appropriately and rigorously?

Reviewer #1: Yes

Reviewer #2: Yes

3. Have the authors made all data underlying the findings in their manuscript fully available (please refer to the Data Availability Statement at the start of the manuscript PDF file)?

Reviewer #1: Yes

Reviewer #2: Yes

4. Is the manuscript presented in an intelligible fashion and written in standard English?

Reviewer #1: Yes

Reviewer #2: No

5. Review Comments to the Author

Reviewer #1: The aim was to assess the extent, content, type, and power of advertising of ultra-processed/unhealthy foods around schools in both urban and rural settings in Kenya. The study presents a detailed methodology using a reference instrument applied worldwide. The results are adequately presented and discussed, with implications for public policy. I suggest adjusting minor points before publication to make the manuscript clearer. Introduction

The introduction is interesting and presents a scenario about advertising ultra-processed foods. However, the justification for the work could be better developed.

Materials and methods

The study counties were selected for convenience, or did you do some random selection?

Line 153: Since you've briefly explained what unprocessed and minimally processed foods are, I suggest you do the same for processed and ultra-processed foods.

Include the software that you used to do the data analysis.

Results

Include country and year in the title tables.

Line 188 – 192: I needed clarification on these two sentences. First, you say that food and beverage advertisements are more common in Nairobi, but in the following sentence, you say that food and beverage products are more advertised in Mombasa. How it's written looks like the same information with different data. What data are you presenting in the first sentence that differs from the data you are presenting in the second sentence?

It would also be interesting to see the information you present in Table 3 stratified by school setting, SES level, and school type (public/private).

Reviewer #2: General comments:

This study assessed the extent, content, type, and power of advertising of ultra-processed/unhealthy foods around schools in both urban and rural settings in Kenya. Overall, this is an interesting study that contributes to the food environment literature in LMICs. The results are as expected with urban areas possessing greater advertisements than rural areas, though there are some key nuances throughout that should be explained.

Specific comments:

- Grammar can be improved throughout

- L60 – “obesogenic environments” is not a standardized term. Would remove this

- Paragraphs 1 and 2 of the introduction are redundant. Can you condense?

- L75 – “HICs” is not defined previously

- L90-92 - The objective statement must be stronger. Why Kenya?

- L115 – “sample size of 500” – specify schools here as it is unclear

- Why wasn’t a listing of schools in Kenya (and your three regions) used, versus an estimation of 500 schools? If this was indeed done, the methods should be rewritten to explain that you started with a listing of schools, stratified by primary, secondary or urban/rural, and you used a prevalence of 23% of unhealthy food advertisements as your outcome in your power calculation.

- Tables 1 and 2 can be condensed into one

- L194 – says you found a higher number of advertisements in private primary, but Table 2 suggests otherwise (public primary is higher)

- Food shops is very vague, and it’s intuitive that more food advertisements would be within/around food shops. Did you collect data on what type of food outlet (i.e., convenience store, sit-down restaurant, fast-food etc.)?

- Discussion should explain nuances in the data such as Baringo country having a 28% increase in UPF advertisements, though having the fewest number of advertisements in total (median of 8 (5,15)) around schools.

- What about policy implications for healthy foods? Your findings reveal that both unprocessed/minimally processed and UPF were similar in % across regions. How can we promote healthy foods?

6. PLOS authors have the option to publish the peer review history of their article (what does this mean? ). If published, this will include your full peer review and any attached files.

**Do you want your identity to be public for this peer review?** For information about this choice, including consent withdrawal, please see our Privacy Policy .

Reviewer #1: No

Reviewer #2: **Yes: ** Bianca Carducci

---

## [Decision Letter · Decision Letter 1]

PGPH-D-24-02106R1

Advertising ultra-processed foods around urban and rural schools in Kenya

Dear Dr. Karugu,

Thank you for submitting your manuscript to PLOS Global Public Health. After careful consideration, we feel that it has merit but does not fully meet PLOS Global Public Health’s publication criteria as it currently stands. Therefore, we invite you to submit a revised version of the manuscript that addresses the points raised during the review process.

We look forward to receiving your revised manuscript.

Kind regards,

Rashmi Josephine Rodrigues, M.D., Ph.D.

Academic Editor

Journal Requirements:

Additional Editor Comments (if provided):

Reviewers' comments:

Reviewer's Responses to Questions

**Comments to the Author**

1. If the authors have adequately addressed your comments raised in a previous round of review and you feel that this manuscript is now acceptable for publication, you may indicate that here to bypass the “Comments to the Author” section, enter your conflict of interest statement in the “Confidential to Editor” section, and submit your "Accept" recommendation.

Reviewer #1: All comments have been addressed

Reviewer #3: All comments have been addressed

2. Does this manuscript meet PLOS Global Public Health’s publication criteria ? Is the manuscript technically sound, and do the data support the conclusions? The manuscript must describe methodologically and ethically rigorous research with conclusions that are appropriately drawn based on the data presented.

Reviewer #1: Yes

Reviewer #3: Yes

3. Has the statistical analysis been performed appropriately and rigorously?

Reviewer #1: Yes

Reviewer #3: Yes

4. Have the authors made all data underlying the findings in their manuscript fully available (please refer to the Data Availability Statement at the start of the manuscript PDF file)?

Reviewer #1: Yes

Reviewer #3: Yes

5. Is the manuscript presented in an intelligible fashion and written in standard English?

Reviewer #1: Yes

Reviewer #3: No

6. Review Comments to the Author

Reviewer #1: The authors presented an improved version of the manuscript, taking into account all the recommended suggestions. I consider the manuscript suitable for publication.

Reviewer #3: At the outset, I must congratulate the authors for the work done which will help policy makers to make better informed decisions and aid in making schools and the environment around them more child friendly and supportive of holistic child growth and development.

General Comments:

1. Good methodology and use of a standard tool which is internationally accepted.

2. Good overall impact and highlight of the ever growing struggle of LES, rural backgrounds and the effect of the Urbanisation.

3. Would suggest the use of a single standard terminology in the study paper - eg: core /non core; healthy/ unhealthy etc.

4. Please re-read the article to screen for grammatical and semantic errors.

5. Please avoid using terminologies such as approximate, about, most / least in the results section of the article since these are definitive values you have obtained through your research, the same can be used in discussion.

6. Could you structure and use bold formatting for your tables to bring out important information, like highlighting the significant p values, etc.

Specific suggestions:

Line 78,79 -Various studies have been conducted in both LMICs and HICs illustrating the extent of marketing "of" health-harming food products to children and adolescents (12–18).

Line 80 - Could you explain the meaning of the phrase "different socioeconomic deprivation areas" and the setting based on the reference provided?

Line 91 - Would suggest revising - "marketing of high foods".

Line 117, 118 - Would suggest revising - "deprived - resource" settings; "margin of error" to Alpha error set to 5%

Line 137 - Delete n - "merchandising) n was also recorded"

Line 138,139, 235 ( table) : Suggest appropriate symbols to be used as of "medium (<1.3...." and "large(2.... " to "medium ( > (more than symbol) 1.3" and "large ( > ( more than symbol) 2...."

Line 143 : Please revise " premium offers were used" as it is not clear.

Line 159 - The reason for additional sub-categorisation as binary variable when an existent NOVA food classification requires explanation and as to how this binary variable was arrived at based on the NOVA food classification which was used.

Line 163 - Would suggest edit " school types and levels and level of" to school types and level with the degree or extent of urbanisation instead.

Line 163 : replace "assess" with "describe", as median and IQRs are not assessment tools, they are more descriptive statistics.

Line 171: Would suggest the mention the tests used for post estimations.

Line 206 / 207 - Mean with a SD above the Mean may not be a great description of the data set. Could consider Median and IQR.

Line 248/249 - It describes conversely results as described, however based on the data, they both highlight that non core or unhealthy foods are more advertised than the other. Also prefer standard terms - either core and non core or healthy and unhealthy.

Line 329 - Could rephrase the sentence " restricted to advertising" to "which are restricted for advertising to children in Kenya"

Line 355 : It is mentioned that "different geographical contexts" are illustrated, however the study is more socioeconomic stratification than the geography.

Line 357 - Would suggest "Better pester power to caregivers to " pester power of children to their caregivers"

Line 359 - Expand NR-NCD.

Looking forward to your edited version. I am confident this paper will do great work in better directing policies and providing direction to the work at hand.

7. PLOS authors have the option to publish the peer review history of their article (what does this mean? ). If published, this will include your full peer review and any attached files.

**Do you want your identity to be public for this peer review?** For information about this choice, including consent withdrawal, please see our Privacy Policy .

Reviewer #1: No

Reviewer #3: **Yes: ** Nikith Austin D'Souza

---

## [Decision Letter · Decision Letter 2]

Advertising ultra-processed foods around urban and rural schools in Kenya

PGPH-D-24-02106R2

Dear Miss Karugu,

We are pleased to inform you that your manuscript 'Advertising ultra-processed foods around urban and rural schools in Kenya' has been provisionally accepted for publication in PLOS Global Public Health.

Best regards,

Rashmi Josephine Rodrigues, M.D., Ph.D.

Academic Editor

Reviewer Comments (if any, and for reference):

Reviewer's Responses to Questions

**Comments to the Author**

1. If the authors have adequately addressed your comments raised in a previous round of review and you feel that this manuscript is now acceptable for publication, you may indicate that here to bypass the “Comments to the Author” section, enter your conflict of interest statement in the “Confidential to Editor” section, and submit your "Accept" recommendation.

Reviewer #3: All comments have been addressed

2. Does this manuscript meet PLOS Global Public Health’s publication criteria ? Is the manuscript technically sound, and do the data support the conclusions? The manuscript must describe methodologically and ethically rigorous research with conclusions that are appropriately drawn based on the data presented.

Reviewer #3: Yes

3. Has the statistical analysis been performed appropriately and rigorously?

Reviewer #3: Yes

4. Have the authors made all data underlying the findings in their manuscript fully available (please refer to the Data Availability Statement at the start of the manuscript PDF file)?

Reviewer #3: Yes

5. Is the manuscript presented in an intelligible fashion and written in standard English?

Reviewer #3: Yes

6. Review Comments to the Author

Reviewer #3: Thank you for accepting the suggestions and reviewers comments.

Good paper! hope it achieves better reforms as hoped for by the research team!

Good luck!

7. PLOS authors have the option to publish the peer review history of their article (what does this mean? ). If published, this will include your full peer review and any attached files.

**Do you want your identity to be public for this peer review?** For information about this choice, including consent withdrawal, please see our Privacy Policy .

Reviewer #3: **Yes: ** Nikith Austin D'Souza MD
